# Spatial Patterns of Macromolecular Composition of Phytoplankton in the Arctic Ocean

Keyseok Choe [1,2], Misun Yun [3,*], Sanghoon Park [1], Eunjin Yang [4], Jinyoung Jung [4], Jaejoong Kang [1], Naeun Jo [1], Jaehong Kim [1], Jaesoon Kim [1] and Sang Heon Lee [1,*]

1 Department of Oceanography, College of Natural Science, Pusan National University, Busan 46241, Korea; keyseok.choe@gmail.com (K.C.); mossinp@pusan.ac.kr (S.P.); jaejung@pusan.ac.kr (J.K.); nadan213@pusan.ac.kr (N.J.); king9527@naver.com (J.K.); jaesoonkim1123@naver.com (J.K.)
2 National Marine Biodiversity Institute of Korea, Seocheon 33662, Korea
3 College of Marine and Environmental Sciences, Tianjin University of Science and Technology, Tianjin 300457, China
4 Korea Polar Research Institute, Incheon 21990, Korea; ejyang@kopri.re.kr (E.Y.); jinyoungjung@kopri.re.kr (J.J.)
* Correspondence: misunyun@tust.edu.cn (M.Y.); sanglee@pusan.ac.kr (S.L.)

**Abstract:** The macromolecular concentrations and compositions of phytoplankton are crucial for the growth or nutritional structure of higher trophic levels through the food web in the ecosystem. To understand variations in macromolecular contents of phytoplankton, we investigated the macromolecular components of phytoplankton and analyzed their spatial pattern on the Chukchi Shelf and the Canada Basin. The carbohydrate (CHO) concentrations on the Chukchi Shelf and the Canada Basin were 50.4–480.8 μg $L^{-1}$ and 35.2–90.1 μg $L^{-1}$, whereas the lipids (LIP) concentrations were 23.7–330.5 μg $L^{-1}$ and 11.7–65.6 μg $L^{-1}$, respectively. The protein (PRT) concentrations were 25.3–258.5 μg $L^{-1}$ on the Chukchi Shelf and 2.4–35.1 μg $L^{-1}$ in the Canada Basin. CHO were the predominant macromolecules, accounting for 42.6% on the Chukchi Shelf and 60.5% in the Canada Basin. LIP and PRT contributed to 29.7% and 27.7% of total macromolecular composition on the Chukchi Shelf and 30.8% and 8.7% in the Canada Basin, respectively. Low PRT concentration and composition in the Canada Basin might be a result from the severe nutrient-deficient conditions during phytoplankton growth. The calculated food material concentrations were 307.8 and 98.9 μg $L^{-1}$, and the average calorie contents of phytoplankton were 1.9 and 0.6 kcal $m^{-3}$ for the Chukchi Shelf and the Canada Basin, respectively, which indicates the phytoplankton on the Chukchi Shelf could provide the large quantity of food material and high calories to the higher trophic levels. Overall, our results highlight that the biochemical compositions of phytoplankton are considerably different in the regions of the Arctic Ocean. More studies on the changes in the biochemical compositions of phytoplankton are still required under future environmental changes.

**Keywords:** macromolecules; phytoplankton; Chukchi Shelf; Canada Basin; food material

## 1. Introduction

The Arctic Ocean is one of the most affected geographical locations in the world due to global climate change. In the Arctic, there has been a rapid decline of sea ice for several decades [1,2], which can be visualized by the downward trend of sea ice range through continuous satellite observations over the past decade [3–5]. In the last 30 years, the density of sea ice has decreased by about 9% every 10 years, and the sea ice thickness has also decreased [6]. With the disappearance of sea ice, various physico-chemical processes are likely to be altered [7–9].

Recent and rapid changes in the marine environment in the Arctic Ocean have been revealed to have significant impacts on the phytoplankton community [10–12]. For example, Kahru et al. [13] revealed that early phytoplankton blooms were caused by a decrease in sea

ice in the Arctic Ocean. According to Ardyna et al. [14], the lengthening of the open water season in the Arctic Ocean was correlated with the increasing occurrence of the autumn bloom. In the areas of the Arctic Ocean where sea ice was absent, satellite observations have shown a significant increase in annual net primary production (NPP) [15–17]. Other than the quantitative changes of phytoplankton, the physiological conditions of phytoplankton appear to be affected by the recent environmental conditions [18–20].

In general, the organic matter produced by phytoplankton is composed of carbohydrates (hereafter, CHO), proteins (hereafter, PRT), and lipids (hereafter, LIP). The composition and synthesis of these major macromolecules of phytoplankton can provide important clues to the physiological status under the environment in which phytoplankton grow since they reflect the rapid adjustment of environmental conditions [21–23]. Furthermore, the relative amount of each macromolecular component in phytoplankton indicates the quality, or nutritional value, of phytoplankton as a food source [24]. The determination of the energy content for phytoplankton can be important since it could be transferred to the marine herbivores and the higher trophic levels and consequently determine the growth of higher trophic levels.

Previously, some studies on the macromolecular composition of phytoplankton were reported to understand their physiological conditions in the Polar Oceans [23,25–32]. The comparison of macromolecular compositions between phytoplankton and microzooplankton was conducted in the Arctic Ocean [27]. Kim et al. [28] revealed that the Antarctic phytoplankton indicated high protein composition under sustained high nutrient conditions, whereas the Arctic phytoplankton produced more lipids [19,20,27] or carbohydrates [30,31]. These studies mainly focused on the macromolecular compositions of or their vertical distributions within the euphotic zone or related environmental factors. Although the macromolecular composition of phytoplankton could be largely affected by environmental conditions in the regions, very little information is available on the spatial pattern of macromolecular concentration, composition, and nutritional value within the regions of the Arctic Ocean.

In this study, we examined the composition and concentration of the macromolecular pool (CHO, PRT, and LIP) of phytoplankton to understand the energy content of Arctic phytoplankton. In addition, we investigated how the spatial variation of the macromolecular composition is linked to physicochemical and biological parameters in different domains. Finally, the energy content of phytoplankton was calculated to estimate the nutritional value, which could be transferred to the organisms in the higher trophic levels in the Arctic ecosystem.

## 2. Materials and Methods

### 2.1. Research Area and Sampling

From 31 July to 23 August 2014, 21 survey stations on the Chukchi Shelf and the Canada Basin were occupied onboard R/V *Araon*. The samples for the macromolecular components were collected from 10 stations of shelf area (hereafter, Chukchi Shelf) and 11 stations of basin area (hereafter, Canada Basin) during the cruise period (Figure 1). The physical data were obtained through CTD (a Sea-Bird 911+) at each station, and seawater samples were obtained by using rosette samplers. A vertical irradiance profile (photosynthesis active irradiance (PAR), 400–700 nm) was obtained using an LI-COR underwater optical sensor mounted on a CTD/rosette sampler to determine the light depth.

### 2.2. Nutrients and Chlorophyll-a Analysis

Seawater samples for nutrient measurements were obtained from different photic depths (100%, 30%, and 1% depths of surface PAR) determined from the underwater PAR sensor. Samples were collected from all of the corresponding light depths, and the surface water collected was used for 100% light treatment. The nutrient concentrations were measured immediately using an automatic nutrient analyzer (SEAL, QuAAtro, Norderstedt, Germany).

Total chlorophyll-a concentrations (Whatman GF/F filter, ø = 24 mm) were analyzed using the method in [33]. The filters (Whatman GF/F filter, ø = 24 mm) were immediately frozen at −80 °C in each petri dish wrapped in aluminum foil until chlorophyll-a extraction at Pusan National University, South Korea. All the samples for chlorophyll-a concentrations were extracted with 90% acetone at −5 °C for 24 h, and the concentrations were measured using a fluorometer (Turner Designs, 10-AU, San Jose, CA, USA), which was calibrated before the analysis.

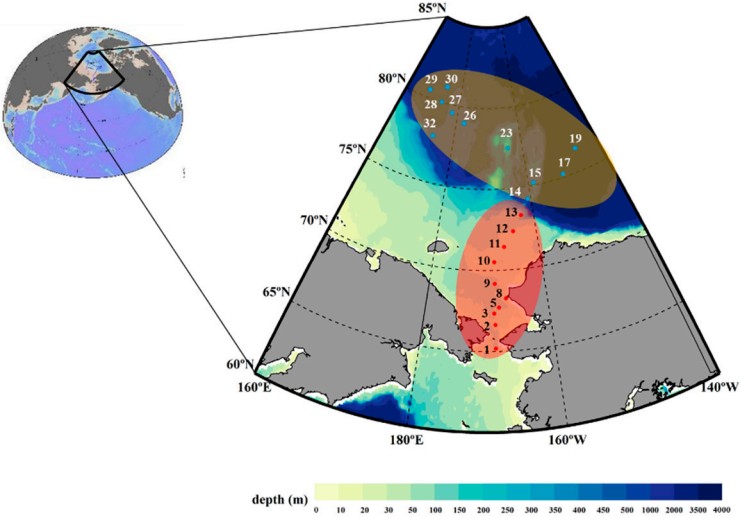

**Figure 1.** Map of the study area with functional regions highlighted in different colors. The red and blue colors indicate the sampling stations of the Chukchi Shelf and Canada Basin, respectively.

### 2.3. Macromolecular Concentration Analysis

Seawater samples (1L) were obtained from 100%, 30%, and 1% depths of surface PAR for the macromolecular concentration in the euphotic zone. The water sample was filtered through a 47-mm Whatman GF/F filter and then immediately stored at −80 °C until further analysis was performed at the home laboratory of Pusan National University. To extract CHO, PRT, and LIP, the phenol-sulfuric acid method [34], modified PRT method [35], and column method [36,37] were used, respectively. The detailed methods are available in [38].

### 2.4. Caloric Content Calculation

The Winberg [24] formula was used to calculate the calorie content (Kcal m$^{-3}$) of food material (FM; the sum of PRT, LIP, and CHO concentrations [39]).

$$\text{Calorie content (Kcal m}^{-3}) = \text{Kcal g FM}^{-1} \times \text{g FM m}^{-3}$$

### 3. Results

#### 3.1. Temperature and Salinity Properties

Figure 2 shows the distributions of water temperature and salinity in the upper ocean. The range of water temperature and salinity of the entire study area varied from −1.8 to 9.7 °C and 26.7 to 32.4 psu, respectively. Regionally, the range of water temperature is from −1.4 to 9.7 °C (1.8 ± 3.4 °C) on the Chukchi Shelf and from −1.8 to 0.2 °C (−0.9 ± 0.5 °C) in the Canada Basin, showing a large difference between the Chukchi Shelf and the Canada Basin. The salinity ranges were 27.2–32.3 psu (30.1 ± 1.9 psu) on the Chukchi Shelf and 26.7–32.4 psu (30.1 ± 1.8 psu) in the Canada Basin, respectively. According to the distribution of salinity, a decreasing tendency was found to the north across the Chukchi Sea (Figure 2).

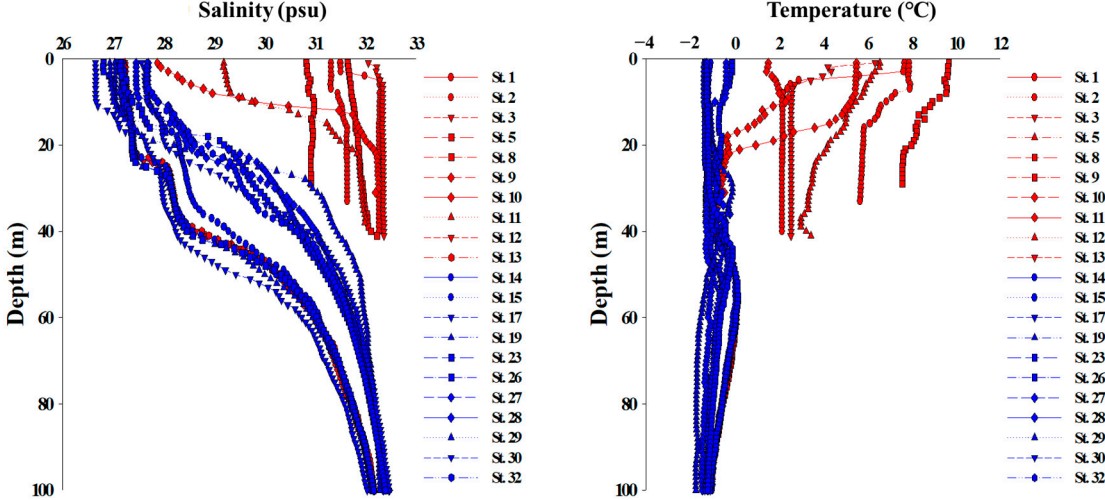

**Figure 2.** Vertical profiles of salinity (**left panel**) and temperature (**right panel**) in the stations of the study area.

### 3.2. Dissolved Inorganic Nutrients

The vertical patterns of dissolved inorganic nutrients showed spatial variations depending on the stations on the Chukchi Shelf, while they were relatively uniform in the Canada Basin (Figure 3). On the Chukchi Shelf, the concentration ranges of $PO_4$, $NO_2 + NO_3$, $NH_4$ and $SiO_2$ were 0.1–2.1, 0–14.3, 0–3.3, and 0.1–51.0 μM, respectively (Figure 3 upper panel). The average concentrations of $PO_4$, $NO_2 + NO_3$, $NH_4$, and $SiO_2$ are $0.7 \pm 0.4$, $1.7 \pm 4.0$, $0.5 \pm 0.8$, and $8.7 \pm 10.1$ μM, respectively. $NO_2 + NO_3$ was depleted at the upper layers on the Chukchi Shelf, and $NH_4$ showed low concentration rather than being depleted. In comparison, the concentration ranges of $PO_4$, $NO_2+NO_3$, $NH_4$, and $SiO_2$ in the Canada Basin were 0.5–1.6, 0–12.3, 0, and 1.6–31.5 μM, respectively (Figure 3 lower panel). The average concentrations of $PO_4$, $NO_2 + NO_3$, $NH_4$, and $SiO_2$ were $0.8 \pm 0.3$, $2.3 \pm 3.8$, $0 \pm 0$, and $7.3 \pm 6.6$ μM, respectively. Similar to what was observed on the Chukchi Shelf, $NO_2+NO_3$ was depleted at the surface of the Canada Basin. Furthermore, $NH_4$ was depleted in the entire water column.

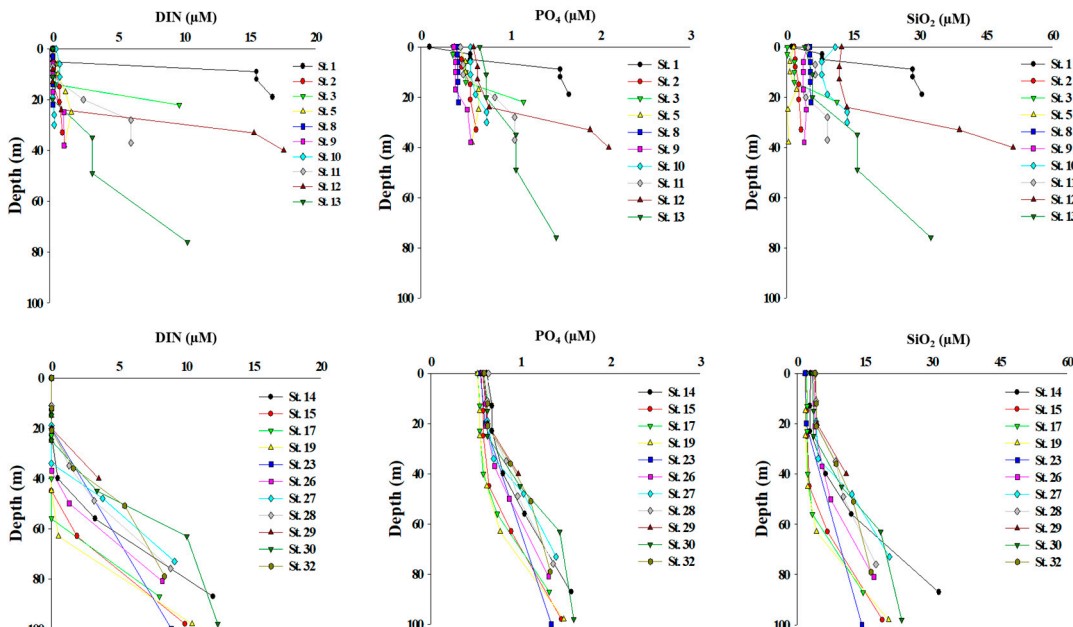

**Figure 3.** Vertical profiles of dissolved inorganic nutrients in stations of the Chukchi Shelf (**upper panel**) and the Canada Basin (**lower panel**).

### 3.3. Total Chlorophyll-a Concentration

During the cruise, the integrated chlorophyll-a concentration from the surface to 1% light depth in the entire study area was $66.3 \pm 84.3$ mg m$^{-2}$, with the total chlorophyll-a concentration showing a large regional variation as it ranged from 5.5 (station 27) to 376.2 mg m$^{-2}$ (station 1) (Figure 4). The average concentration of chlorophyll-a on the Chukchi Shelf was $98.6 \pm 104.3$ mg m$^{-2}$, which was approximately three times that ($30.3 \pm 31.5$ mg m$^{-2}$) in the Canada Basin.

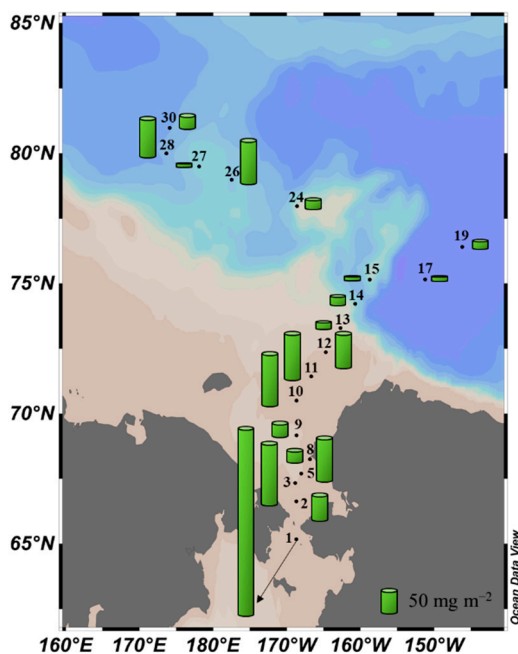

**Figure 4.** Spatial distribution of the chlorophyll-a concentration integrated from the surface to 1% light depth.

### 3.4. Vertical Distribution of Macromolecular Concentration and Composition on the Chukchi Shelf

Quantitative concentrations and relative ratios of CHO, PRT, and LIP on the Chukchi Shelf are summarized in Tables 1–3. The range of CHO, PRT, and LIP concentrations of phytoplankton was 48.2–409.3, 35.4–123.3, and 25.7–326.7 µg L$^{-1}$, respectively, at the surface of the Chukchi Shelf (Table 1). The average CHO concentration at each station was $114.7 \pm 106.8$ µg L$^{-1}$ with a 43.6% contribution, being the dominant macromolecule found in phytoplankton. LIP ($86.4 \pm 93.1$ µg L$^{-1}$) and PRT ($60.2 \pm 26.5$ µg L$^{-1}$) contributed to 29.3% and 27.1% of the total compositions, respectively. The range of FM concentration was 117.6–859.3 µg L$^{-1}$ ($261.3 \pm 221.2$ µg L$^{-1}$). At 30% light depth (Table 2), the range of CHO concentration was determined to be 43.8–919.4 µg L$^{-1}$ ($167.3 \pm 267.9$ µg L$^{-1}$), in which the macromolecule accounted for 42.4% of the total composition. The range and contribution of LIP concentration were 19.2–546.4 µg L$^{-1}$ ($116.5 \pm 161.3$ µg L$^{-1}$) and 30.1%, respectively, and the range and contribution of PRT concentration were 27.2–586.8 µg L$^{-1}$ ($107.6 \pm 170.8$ µg L$^{-1}$) and 27.5%, respectively. Both the average concentration and contribution of LIP and PRT tended to increase compared to those at the surface layer. The range of FM concentration was 90.1–2052.6 µg L$^{-1}$ ($391.5 \pm 597.0$ µg L$^{-1}$), which was considerably higher than that at the surface. Compared to 30% light depth, the average composition of CHO concentration was slightly reduced to 41.7%, and its range was 34.9–259.0 µg L$^{-1}$ ($101.4 \pm 64.0$ µg L$^{-1}$) at 1% euphotic depth (Table 3). At this depth, the ranges of LIP and PRT concentrations were found to be 12.0–256.7 ($90.5 \pm 79.7$ µg L$^{-1}$) and 10.0–185.8 µg L$^{-1}$ ($78.7 \pm 52.8$ µg L$^{-1}$). Consequently, the LIP contributed to 29.7% of the overall composition while PRT contributed to 28.6%, respectively. The CHO synthesis continued to decrease, whereas the PRT synthesis increased with depth. The range of FM concentration was 56.9–701.6 µg L$^{-1}$ ($270.6 \pm 183.2$ µg L$^{-1}$) at the 1% depth.

**Table 1.** Concentrations and compositions of macromolecular components (carbohydrates: CHO, proteins: PRT, and lipids: LIP), food material (FM), and calorie content of phytoplankton at 100% light depth on the Chukchi Shelf.

| Station | Light Depth (%) | CHO ($\mu g\ L^{-1}$) | PRT ($\mu g\ L^{-1}$) | LIP ($\mu g\ L^{-1}$) | FM ($\mu g\ L^{-1}$) | CHO/FM (%) | PRT/FM (%) | LIP/FM (%) | Calorific Content (Kcal $m^{-3}$) |
|---|---|---|---|---|---|---|---|---|---|
| 1 | 100 | 409.3 | 123.3 | 326.7 | 859.3 | 47.6 | 14.3 | 38.0 | 5.5 |
| 2 | 100 | 80.1 | 54.3 | 135.9 | 270.3 | 29.6 | 20.1 | 50.3 | 1.9 |
| 3 | 100 | 139.1 | 77.6 | 123.9 | 340.6 | 40.8 | 22.8 | 36.4 | 2.2 |
| 5 | 100 | 62.0 | 67.5 | 43.0 | 172.6 | 35.9 | 39.1 | 24.9 | 1.0 |
| 8 | 100 | 113.6 | 70.0 | 53.8 | 237.4 | 47.8 | 29.5 | 22.6 | 1.4 |
| 9 | 100 | 65.6 | 38.2 | 29.5 | 133.3 | 49.2 | 28.7 | 22.1 | 0.8 |
| 10 | 100 | 74.0 | 40.4 | 26.9 | 141.3 | 52.4 | 28.6 | 19.1 | 0.8 |
| 11 | 100 | 48.2 | 43.6 | 25.7 | 117.6 | 41.0 | 37.1 | 21.9 | 0.7 |
| 12 | 100 | 82.3 | 51.5 | 40.1 | 173.9 | 47.4 | 29.6 | 23.0 | 1.0 |
| 13 | 100 | 73.1 | 35.4 | 58.1 | 166.7 | 43.9 | 21.2 | 34.9 | 1.0 |
| Average | | 114.7 | 60.2 | 86.4 | 261.3 | 43.6 | 27.1 | 29.3 | 1.6 |
| SD | | 106.8 | 26.5 | 93.1 | 221.2 | 6.9 | 7.6 | 10.1 | 1.4 |

**Table 2.** Concentrations and compositions of macromolecular components (carbohydrates: CHO, proteins: PRT, and lipids: LIP), food material (FM), and calorie content of phytoplankton at 30% light depth on the Chukchi Shelf.

| Station | Light Depth (%) | CHO ($\mu g\ L^{-1}$) | PRT ($\mu g\ L^{-1}$) | LIP ($\mu g\ L^{-1}$) | FM ($\mu g\ L^{-1}$) | CHO/FM (%) | PRT/FM (%) | LIP/FM (%) | Calorific Content (Kcal $m^{-3}$) |
|---|---|---|---|---|---|---|---|---|---|
| 1 | 30 | 919.4 | 586.8 | 546.4 | 2052.6 | 44.8 | 28.6 | 26.6 | 12.2 |
| 2 | 30 | 88.9 | 62.5 | 54.2 | 205.7 | 43.2 | 30.4 | 26.3 | 1.2 |
| 3 | 30 | 199.7 | 126.2 | 215.4 | 541.3 | 36.9 | 23.3 | 39.8 | 3.6 |
| 5 | 30 | 78.9 | 62.2 | 76.2 | 217.3 | 36.3 | 28.6 | 35.1 | 1.4 |
| 8 | 30 | 96.8 | 67.2 | 45.0 | 209.1 | 46.3 | 32.1 | 21.5 | 1.2 |
| 9 | 30 | 53.7 | 40.4 | 22.8 | 116.8 | 45.9 | 34.6 | 19.5 | 0.7 |
| 10 | 30 | 59.6 | 36.5 | 28.3 | 124.4 | 47.9 | 29.3 | 22.8 | 0.7 |
| 11 | 30 | 43.8 | 27.2 | 19.2 | 90.1 | 48.6 | 30.1 | 21.3 | 0.5 |
| 12 | 30 | 70.2 | 37.2 | 74.4 | 181.8 | 38.6 | 20.4 | 40.9 | 1.2 |
| 13 | 30 | 62.3 | 30.4 | 83.4 | 176.1 | 35.4 | 17.3 | 47.4 | 1.2 |
| Average | | 167.3 | 107.6 | 116.5 | 391.5 | 42.4 | 27.5 | 30.1 | 2.4 |
| SD | | 267.9 | 170.8 | 161.3 | 597.0 | 5.1 | 5.4 | 9.9 | 3.5 |

**Table 3.** Concentrations and compositions of macromolecular components (carbohydrates: CHO, proteins: PRT, and lipids: LIP), food material (FM), and calorie content of phytoplankton at 1% light depth on the Chukchi Shelf.

| Station | Light Depth (%) | CHO ($\mu g\ L^{-1}$) | PRT ($\mu g\ L^{-1}$) | LIP ($\mu g\ L^{-1}$) | FM ($\mu g\ L^{-1}$) | CHO/FM (%) | PRT/FM (%) | LIP/FM (%) | Calorific Content (Kcal $m^{-3}$) |
|---|---|---|---|---|---|---|---|---|---|
| 1 | 1 | 113.6 | 65.4 | 118.4 | 297.4 | 38.2 | 22.0 | 39.8 | 2.0 |
| 2 | 1 | 73.8 | 74.3 | 158.3 | 306.5 | 24.1 | 24.3 | 51.7 | 2.2 |
| 3 | 1 | 92.0 | 144.0 | 157.9 | 394.0 | 23.4 | 36.6 | 40.1 | 2.7 |
| 5 | 1 | 65.9 | 57.9 | 36.1 | 159.9 | 41.2 | 36.2 | 22.6 | 0.9 |
| 8 | 1 | 77.3 | 60.8 | 36.7 | 174.7 | 44.2 | 34.8 | 21.0 | 1.0 |
| 9 | 1 | 85.0 | 30.4 | 18.8 | 134.2 | 63.4 | 22.6 | 14.0 | 0.7 |
| 10 | 1 | 259.0 | 185.8 | 256.7 | 701.6 | 36.9 | 26.5 | 36.6 | 4.5 |
| 11 | 1 | 59.1 | 52.9 | 52.6 | 164.6 | 35.9 | 32.1 | 31.9 | 1.0 |
| 12 | 1 | 152.9 | 105.1 | 57.7 | 315.7 | 48.4 | 33.3 | 18.3 | 1.8 |
| 13 | 1 | 34.9 | 10.0 | 12.0 | 56.9 | 61.3 | 17.6 | 21.1 | 0.3 |
| Average | | 101.4 | 78.7 | 90.5 | 270.6 | 41.7 | 28.6 | 29.7 | 1.7 |
| SD | | 64.0 | 52.8 | 79.7 | 183.2 | 13.4 | 6.8 | 12.1 | 1.2 |

*3.5. Vertical Distribution of Macromolecular Concentration and Composition in the Canada Basin*

The range and contribution of CHO concentration of phytoplankton at the surface layer in the Canada Basin were dominant, being 28.8–177.2 µg L$^{-1}$ (70.9 ± 42.3 µg L$^{-1}$) and 61.9%, respectively (Table 4). LIP concentration contributed to 29.7% of the total composition with a concentration range of 10.9–81.0 µg L$^{-1}$ (34.2 ± 26.3 µg L$^{-1}$), and the range of PRT concentration was 3.2–13.6 µg L$^{-1}$ (8.0 ± 3.2 µg L$^{-1}$) with a contribution of 8.4% at the surface layer. The range of FM concentration was 55.5–250.2 µg L$^{-1}$ (113.1 ± 51.5 µg L$^{-1}$). At 30% euphotic depth (Table 5), CHO contributed to 57.6% of the total macromolecular composition and had a concentration range of 35.0–77.1 µg L$^{-1}$ (48.2 ± 13.1 µg L$^{-1}$). Compared to the surface layer, the concentration of CHO was significantly reduced, and its contribution was somewhat decreased. On the other hand, the contribution of LIP concentration was 32.7%, which was increased compared to that at the surface, with a range of 10.4–58.7 µg L$^{-1}$. However, the average concentration of LIP was reduced to 29.3 µg L$^{-1}$ (±18.7 µg L$^{-1}$) and smaller compared to that at the surface. The range and content of PRT concentration were 2.9–15.4 µg L$^{-1}$ (8.4 ± 3.8 µg L$^{-1}$) and 9.8%, respectively, and it was identified to be greater than what was found at the surface. The range of FM concentration was between 56.0 and 110.0 µg L$^{-1}$ (85.9 ± 19.1 µg L$^{-1}$). At 1% light depth (Table 6), both the range (33.3–92.9 µg L$^{-1}$) and content (62.2%) of CHO concentrations increased. The range and contribution of LIP concentration were determined to be 9.0–92.5 µg L$^{-1}$ (32.2 ± 28.6 µg L$^{-1}$) and 29.9%, respectively. A decrease in the range and contribution of PRT concentration was observed. The range of PRT concentration was 0.7–83.8 µg L$^{-1}$ (11.4 ± 24.1 µg L$^{-1}$) with a total contribution of 7.9% and the range of FM concentration was 46.8–269.2 µg L$^{-1}$ (97.5 ± 63.1 µg L$^{-1}$) at 1% light depth in the Canada Basin.

**Table 4.** Concentrations and compositions of macromolecular components (CHO, PRT, and LIP), food material (FM), and calorie content of phytoplankton at 100% light depth in the Canada Basin.

| Station | Light Depth (%) | CHO (µg L$^{-1}$) | PRT (µg L$^{-1}$) | LIP (µg L$^{-1}$) | FM (µg L$^{-1}$) | CHO/FM (%) | PRT/FM (%) | LIP/FM (%) | Calorific Content (Kcal m$^{-3}$) |
|---|---|---|---|---|---|---|---|---|---|
| 14 | 100 | 51.1 | 13.6 | 14.4 | 79.1 | 64.6 | 17.2 | 18.2 | 0.4 |
| 15 | 100 | 177.2 | 3.6 | 69.5 | 250.2 | 70.8 | 1.4 | 27.8 | 1.4 |
| 17 | 100 | 28.8 | 3.2 | 66.1 | 98.1 | 29.3 | 3.3 | 67.4 | 0.8 |
| 19 | 100 | 55.7 | 10.0 | 81.0 | 146.7 | 38.0 | 6.8 | 55.2 | 1.1 |
| 23 | 100 | 50.1 | 6.4 | 24.0 | 80.5 | 62.3 | 8.0 | 29.7 | 0.5 |
| 26 | 100 | 98.7 | 4.7 | 17.4 | 120.8 | 81.7 | 3.9 | 14.4 | 0.6 |
| 27 | 100 | 83.2 | 9.8 | 17.0 | 110.0 | 75.6 | 8.9 | 15.5 | 0.6 |
| 28 | 100 | 35.9 | 8.7 | 10.9 | 55.5 | 64.7 | 15.7 | 19.6 | 0.3 |
| 29 | 100 | 93.1 | 9.4 | 12.7 | 115.2 | 80.8 | 8.2 | 11.1 | 0.6 |
| 30 | 100 | 69.7 | 7.3 | 18.5 | 95.4 | 73.0 | 7.6 | 19.4 | 0.5 |
| 32 | 100 | 37.1 | 10.9 | 45.1 | 93.1 | 39.9 | 11.7 | 48.4 | 0.6 |
| Average | | 70.9 | 8.0 | 34.2 | 113.1 | 61.9 | 8.4 | 29.7 | 0.7 |
| SD | | 42.3 | 3.2 | 26.3 | 51.5 | 18.1 | 4.9 | 18.8 | 0.3 |

**Table 5.** Concentrations and compositions of macromolecular components (CHO, PRT, and LIP), food material (FM), and calorie content of phytoplankton at 30% light depth in the Canada Basin.

| Station | Light Depth (%) | CHO (µg L$^{-1}$) | PRT (µg L$^{-1}$) | LIP (µg L$^{-1}$) | FM (µg L$^{-1}$) | CHO/FM (%) | PRT/FM (%) | LIP/FM (%) | Calorific Content (Kcal m$^{-3}$) |
|---|---|---|---|---|---|---|---|---|---|
| 14 | 30 | 36.4 | 6.8 | 55.4 | 98.6 | 37.0 | 6.9 | 56.1 | 0.7 |
| 15 | 30 | 47.3 | 2.9 | 14.6 | 64.8 | 73.0 | 4.4 | 22.6 | 0.3 |
| 17 | 30 | 35.0 | 5.0 | 58.7 | 98.8 | 35.5 | 5.1 | 59.5 | 0.7 |
| 19 | 30 | 54.7 | 15.4 | 12.6 | 82.7 | 66.2 | 18.6 | 15.3 | 0.4 |
| 23 | 30 | 39.9 | 5.7 | 10.4 | 56.0 | 71.2 | 10.2 | 18.6 | 0.3 |

**Table 5.** *Cont.*

| Station | Light Depth (%) | CHO (µg L$^{-1}$) | PRT (µg L$^{-1}$) | LIP (µg L$^{-1}$) | FM (µg L$^{-1}$) | CHO/FM (%) | PRT/FM (%) | LIP/FM (%) | Calorific Content (Kcal m$^{-3}$) |
|---|---|---|---|---|---|---|---|---|---|
| 26 | 30 | 38.7 | 4.7 | 32.6 | 76.0 | 50.9 | 6.2 | 42.9 | 0.5 |
| 27 | 30 | 60.7 | 10.5 | 28.2 | 99.4 | 61.1 | 10.6 | 28.3 | 0.6 |
| 28 | 30 | 36.6 | 7.6 | 14.8 | 59.0 | 62.0 | 12.9 | 25.0 | 0.3 |
| 29 | 30 | 77.1 | 12.0 | 12.5 | 101.7 | 75.9 | 11.8 | 12.3 | 0.5 |
| 30 | 30 | 56.9 | 11.6 | 29.3 | 97.8 | 58.2 | 11.9 | 29.9 | 0.6 |
| 32 | 30 | 46.6 | 9.8 | 53.6 | 110.0 | 42.4 | 8.9 | 48.7 | 0.8 |
| Average | | 48.2 | 8.4 | 29.3 | 85.9 | 57.6 | 9.8 | 32.7 | 0.5 |
| SD | | 13.1 | 3.8 | 18.7 | 19.1 | 14.3 | 4.1 | 16.5 | 0.2 |

**Table 6.** Concentrations and compositions of macromolecular components (CHO, PRT, and LIP), food material (FM), and calorie content of phytoplankton at 1% light depth in the Canada Basin.

| Station | Light Depth (%) | CHO (µg L$^{-1}$) | PRT (µg L$^{-1}$) | LIP (µg L$^{-1}$) | FM (µg L$^{-1}$) | CHO/FM (%) | PRT/FM (%) | LIP/FM (%) | Calorific Content (Kcal m$^{-3}$) |
|---|---|---|---|---|---|---|---|---|---|
| 14 | 1 | 38.7 | 3.6 | 24.4 | 66.6 | 58.1 | 5.4 | 36.6 | 0.4 |
| 15 | 1 | 45.9 | 0.7 | 18.2 | 64.8 | 70.8 | 1.1 | 28.1 | 0.4 |
| 17 | 1 | 41.8 | 4.6 | 72.1 | 118.5 | 35.2 | 3.9 | 60.8 | 0.9 |
| 19 | 1 | 47.7 | 8.2 | 9.0 | 64.9 | 73.4 | 12.7 | 13.9 | 0.3 |
| 23 | 1 | 70.2 | 4.6 | 57.0 | 131.8 | 53.3 | 3.5 | 43.2 | 0.9 |
| 26 | 1 | 81.0 | 2.9 | 22.6 | 106.5 | 76.1 | 2.7 | 21.2 | 0.6 |
| 27 | 1 | 33.3 | 2.5 | 10.9 | 46.8 | 71.3 | 5.4 | 23.3 | 0.3 |
| 28 | 1 | 55.3 | 6.5 | 9.4 | 71.2 | 77.6 | 9.2 | 13.2 | 0.4 |
| 29 | 1 | 92.9 | 83.8 | 92.5 | 269.2 | 34.5 | 31.1 | 34.3 | 1.7 |
| 30 | 1 | 39.1 | 3.6 | 10.1 | 52.9 | 74.0 | 6.9 | 19.2 | 0.3 |
| 32 | 1 | 47.3 | 4.4 | 28.0 | 79.7 | 59.4 | 5.5 | 35.1 | 0.5 |
| Average | | 53.9 | 11.4 | 32.2 | 97.5 | 62.2 | 7.9 | 29.9 | 0.6 |
| SD | | 19.2 | 24.1 | 28.6 | 63.1 | 15.7 | 8.3 | 14.1 | 0.4 |

### *3.6. Temperature, Salinity, Nutrients, and Macromolecular Concentration along the Shelf/Basin Gradient*

The transect from station 10 to station 15 was run to understand the variability of environmental conditions and macromolecular concentrations along the shelf/basin gradient (Figure 5). Due to sea ice, the water column was in a freezing condition toward the basin. Based on the salinity and temperature distributions, station 12 was determined as the sea-ice edge. The dissolved inorganic nutrients at the depths around the sea-ice edge were distinctly high and decreased toward the basin. The chlorophyll-a distribution gradually decreased toward the basin. The chlorophyll-a concentration at the shelf was lowest at the surface and increased with depth.

The highest CHO concentration was observed at station 10 (>200 µg L$^{-1}$). Similarly, the LIP was highest at station 10. The upper layer of the basin showed a high LIP concentration. The distribution of PRT concentration was also similar to those of CHO or LIP, but PRT concentration at the ice-covered stations of the basin (from station 13 to station 15) was distinctly low. Interestingly, CHO and PRT concentrations were two or three times lower at the stations of the basin compared to the shelf area.

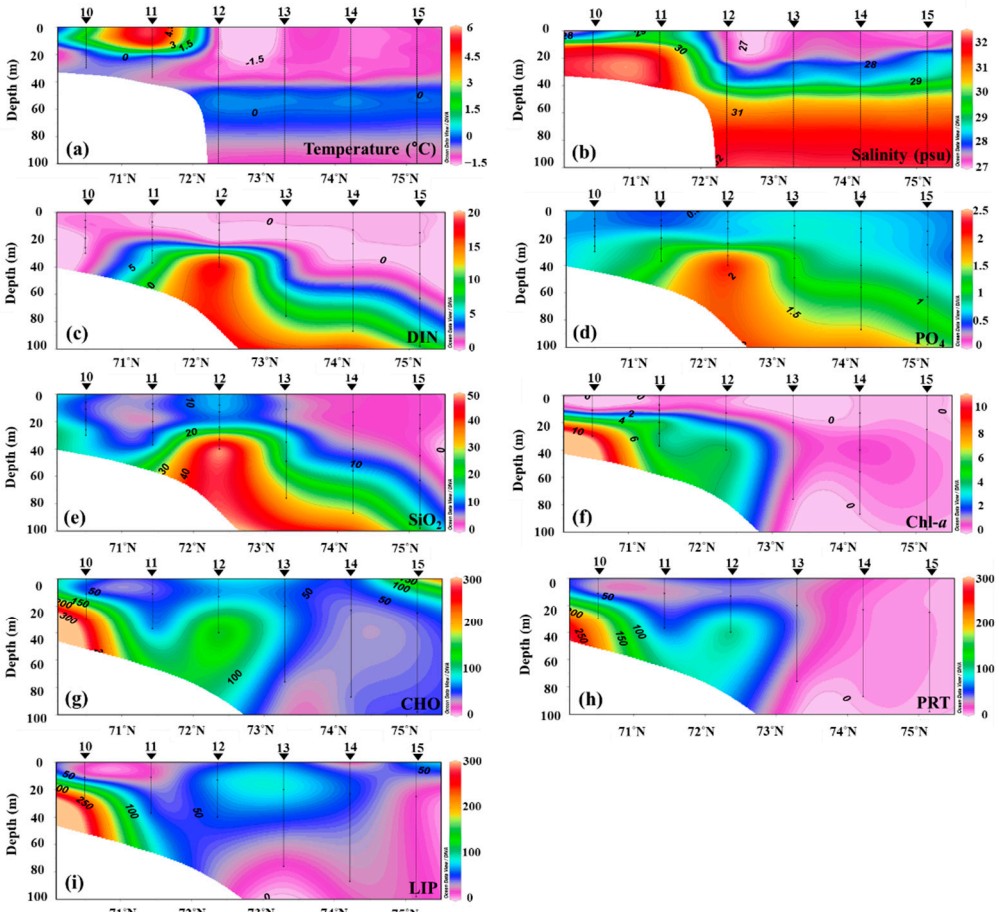

**Figure 5.** The temperature (**a**), salinity (**b**), DIN concentration (**c**), PO$_4$ concentration (**d**), SiO$_2$ concentration (**e**), chlorophyll-a concentration (**f**), CHO concentration (**g**), PRT concentration (**h**), and LIP concentration (**i**) distributions in the vertical section from the Chukchi shelf to Canada Basin in the Arctic Ocean.

*3.7. Spatial Distribution of the Macromolecular Composition*

No statistically significant difference in the relative percentage of each macromolecular component was found among the different light depths (*t*-test, *p* > 0.05). Thus, each component was averaged from three different light depths. Figure 6 shows a spatial distribution of the macromolecular composition over the euphotic zone during this study period. The contribution of CHO was the lowest at station 2 and highest at station 15. The LIP component accounted for 18.5–62.6% of the total macromolecular composition among the stations. The PRT contributed to 2.3–34.6% of the total macromolecular composition. Over the entire study area, CHO was identified as the biggest contributor (52.0%) to the overall average composition of phytoplankton, which was followed by LIP (30.3%) and PRT (17.8%). Regionally, the CHO contributed to 42.6% and 60.5% of the total macromolecular compositions for the Chukchi Shelf and the Canada Basin, respectively. The LIP contributions were 29.7% on the Chukchi Shelf and 30.8% in the Canada Basin, respectively. The average PRT composition was significantly low (*t*-test, *p* < 0.05) in the Canada Basin (8.7% of total) compared to the Chukchi Shelf (27.7% of total) (Figure 6).

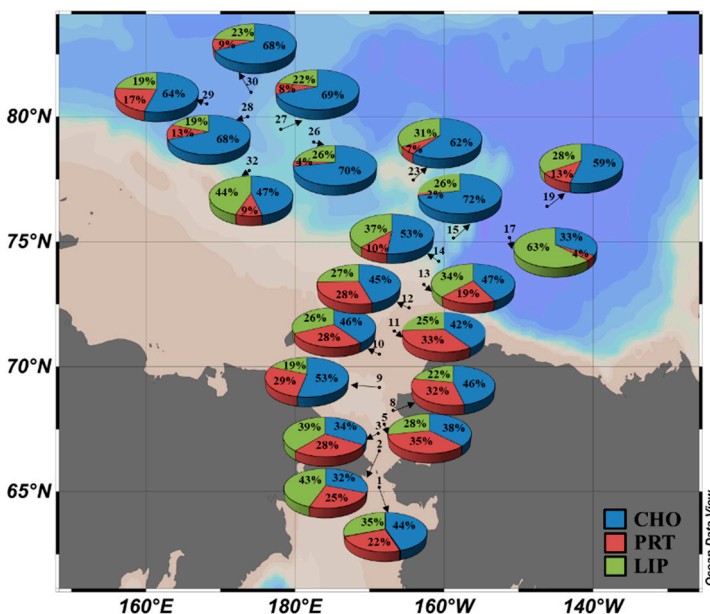

**Figure 6.** Spatial distribution of the macromolecular compositions of phytoplankton during the 2014 expedition.

## 4. Discussion

### 4.1. Major Controlling Factors for the Spatial Variation in Macromolecular Composition

In this study, the major macromolecule contributing to the overall average composition of phytoplankton was determined to be CHO on the Chukchi Shelf and the Canada Basin (Figure 7). It is interesting compared to those previously reported from the other regions of the polar oceans. For example, Yun et al. [27] observed a higher rate of LIP (58%) compared to CHO or PRT in the phytoplankton in the northern Chukchi Sea. In the Antarctic Ocean, Fabiano et al. [40,41] reported a high contribution, of 50% or more, of PRT to the total FM. Kim et al. [28] also found the high contribution of PRT (67%) to the macromolecular composition of phytoplankton in the Amundsen Sea due to the high concentrations of nitrate + nitrite. In general, the composition of PRT in the phytoplankton increases under nitrogen saturation conditions [40,42]. When the nitrogen or phosphorus is limited, triglycerides, which are energy stores, increase and are converted from PRT metabolism to LIP or CHO metabolisms [23,43,44]. Since CHO and LIP are not nitrogen-derived substrates, the accumulation of these storage compounds can be a reaction mechanism under nitrogen-deficient conditions [45]. In particular, LIP acts as a secondary storage material for the survival of long-term nitrogen conditions due to the fat synthetase system [46]. Thus, photosynthesis products are converted from CHO to LIP synthesis in a nitrogen-depleted environmental condition for an extended period [45,46], even though preferred accumulation in CHO or LIP compounds as a reservoir appears to be specific to species [45]. Some oily diatom species assimilate LIP as a major storage component under nitrogen or silicon restrictions [47,48]. Indeed, Ahn et al. [31] observed a sharp increase in LIP concentration with an increase in micro-phytoplankton in the Arctic Ocean. However, Harrison et al. [49] and Wear et al. [50] reported that diatoms have a relatively constant LIP under nitrogen deficiency while rapidly increasing CHO content and decreasing PRT. Therefore, the high CHO and moderate LIP compositions in the present study might have been due to the deficient nutrient conditions, which is consistent with previous findings [49,51], while PRT production predominates under nitrogen-rich conditions [40,42]. Consequently, it implies that major inorganic nutrients, especially nitrogen, are important controlling factors for the macromolecular composition of phytoplankton.

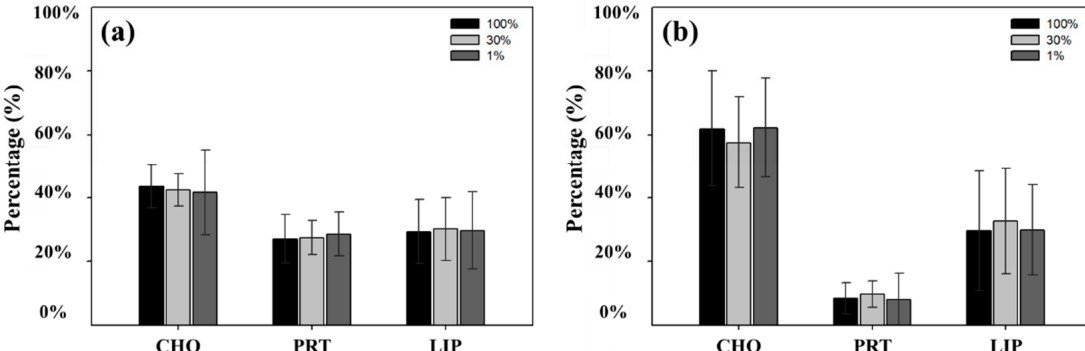

**Figure 7.** Contribution of each macromolecular component at three light depths on the Chukchi Shelf (**a**) and in the Canada Basin (**b**).

During the cruise, the phytoplankton in the Canada Basin were observed to have a significantly lower PRT (8.7%) compared to that of the Chukchi Shelf. This might be related to different conditions of nutrient limitation. Normally, nutrient deficiency can be indicated following as; nitrogen limit condition with N/P ratio with < 10 and Si/N ratio > 1, phosphorus restriction under N/P ratio > 22 and Si/P ratio > 22, and silicon limitation in Si/N ratio < 1 and Si/P ratio < 10 [52]. In terms of macromolecular composition, values with a higher PRT/CHO ratio (>1) are observed in the areas with high productivity or nitrogen-rich blooms [40], while the lower ratios (<1) are in nitrogen deficiency conditions [39]. Thus, low N/P ratios, PRT/CHO ratios, and PRT/LIP ratios indicate that nitrogen is particularly limited in the environment. In this study, the average molar ratio of $(NO_3 + NO_2 + NH_4):PO_4$ and $SiO_2:(NO_3 + NO_2 + NH_4)$ in the euphotic zone was 3.0:1 and 3.9:1 on the Chukchi Shelf, respectively, and 3.4:1 and 2.8:1 in the Canada Basin, respectively. Overall, the ratio of N/P was significantly lower than the Redfield ratio [53]. The PRT/CHO ratio and PRT/LIP ratio of the Canada Basin are 0.16 and 0.33, respectively, which are significantly lower (*t*-test, $p < 0.05$) than the PRT/CHO ratio (0.68) and PRT/LIP ratio (1) of the Chukchi Shelf. These results elucidate that the deficiency of nitrogen in the study area was present during the cruise, and it was especially noticeable that the nitrogen utilization by the phytoplankton in the Canada Basin was severely restricted. Therefore, the substantially low PRT composition of phytoplankton in the Canada Basin could be caused by the result of severe nitrogen deficiency during phytoplankton growth.

According to Suárez et al. [54], the light condition could act as an important factor for determining different macromolecular compositions of phytoplankton. The macromolecular composition of phytoplankton can vary depending on the amount of light [55,56]. For example, an increase in PRT is observed with a decrease in light intensity because of the lower illuminance saturation level of PRT in comparison to other macromolecules [54,57]. In contrast, the productions of CHO and LIP as storage materials can be observed under an excessive energy supply condition [54,58]. Suárez et al. [54] also reported that lower irradiance was more relevant in PRT synthesis than in LIP synthesis. However, no distinct pattern of macromolecular compositions was observed among three different light depths in this study (mentioned in Section 3.7), although the relatively higher protein concentrations were observed at deeper depths (30% and 1% light levels) than at the surface. In addition, the stations in the ice-covered Canada Basin showed significantly low PRT composition, even though it was thought to be a low light condition (Figure 5). Thus, we could conclude that the light condition might be insignificant in controlling the macromolecular composition during this study period.

*4.2. The Implication of Macromolecular Composition as Energy Content Aspect*

In this study, the concentration ranges of CHO, LIP, and PRT were substantially higher on the Chukchi Shelf than in the Canada Basin (Table 7). As a result, the average FMs were 307.8 µg L$^{-1}$ on the Chukchi Shelf and 98.9 µg L$^{-1}$ in the Canada Basin, respectively. The average calorie content of phytoplankton for the Chukchi Shelf and the Canada Basin

during the cruise was 1.9 kcal m$^{-3}$ and 0.6 kcal m$^{-3}$, respectively (Table 7). The overall FM concentration and calorie content of phytoplankton were three times higher on the Chukchi Shelf than in the Canada Basin, which implies that the phytoplankton on the Chukchi Shelf could provide higher FM and calories to the upper trophic levels in the Arctic ecosystem.

According to previous studies, the average calorie contents were 1.0 and 1.2 kcal m$^{-3}$ in the northern Chukchi Sea of the Arctic Ocean [59,60]. Fabiano et al. [41] reported the calorie content of 1.6 kcal m$^{-3}$ in the Ross Sea of the Antarctic Ocean. Recently, Kim et al. [32] observed the different calorie contents of phytoplankton between the two different periods in the Ross Sea of the Antarctic Ocean, indicating 1.3 kcal m$^{-3}$ during the ice-free period and 0.6 kcal m$^{-3}$ during the ice-covered season. Although Kim et al. [29] reported the exceptionally high-calorie content in the productive polynyas of the Amundsen Sea, the calorie content from our study is in a similar range with the previous studies in the polar oceans (Table 7). According to Kim et al. [32], the PRT concentration during the ice-free period in the Ross Sea of the Antarctic Ocean was 20 times increased than that during the ice-covered period, even though CHO or LIP concentrations showed a slight increase (Table 7). If it can be applied in the Arctic Ocean, the PRT concentration than other components might be largely increased under a decrease in sea ice conditions in the Arctic Ocean. Subsequently, the PRT composition that predominates as sea ice decreases could lead to a potential change in the calorie content from an energy point of view. In particular, the macromolecular composition or calorie content of the phytoplankton in ice-covered regions, such as the Canada Basin, might be anticipated to be changed. Consequently, the rich protein-containing FM might be transferred to the upper trophic levels under ongoing and future sea-ice decrease conditions. Thus, the potential effects of the different macromolecular compositions of phytoplankton on the upper trophic levels need to be further evaluated. Above all, in terms of sea ice change, the variability of macromolecular concentration, composition, and calorie content could be important in the Arctic Ocean under ongoing environmental changes.

Table 7. Comparison of carbohydrates (CHO), proteins (PRT), lipids (LIP), food material (FM) concentrations, and calorie content of phytoplankton at different regions of the Polar Oceans. Given were range or mean values.

| Region | Season (Period) | CHO (µg L$^{-1}$) | PRT (µg L$^{-1}$) | LIP (µg L$^{-1}$) | FM (µg L$^{-1}$) | Caloric Content (kcal m$^{-3}$) | References |
|---|---|---|---|---|---|---|---|
| Northern Chukchi Sea | 30 July–19 August 2011 | 21.8–146.7 | 0.7–86.3 | 50.2–105.0 | 149.2 ± 36.5 | 1.0 ± 0.2 | Kim et al. (2015) [59] |
| Northern Chukchi Sea | 1 August–10 September 2012 | 15.9–88.0 | 9.2–183.1 | 37.0–147.4 | 156.4 | 1.2 ± 0.2 | Yun et al. (2015) [27] |
| Chukchi Sea | 7–24 August 2017 | 29.9–406.4 | 9.7–573.8 | 5.4–169.1 | 180.5 ± 195.3 | - | Kim et al. (2020) [61] |
| Laptev and East Siberian Seas | 21 August–22 September 2013 | 29–161 | 22–132 | 15–71 | - | - | Ahn et al. (2019) [30] |
| Northern Kara Sea | | 45.9–67.7 | 22.0–50. 8 | 15.4–44.0 | - | - | |
| Laptev Sea | 18 August–30 September 2015 | 44.4–72.2 | 9.8–22.0 | 20.3–37.1 | - | - | Ahn et al. (2020) [31] |
| Western East Siberian Sea | | 55.7–115. 5 | 1.7–30. 5 | 24.7–67.6 | - | - | |
| Amundsen Sea | 11 February–14 March 2012 | 2.8–216.0 | 5.9–396.2 | 13.2–36.9 | 219.4 ± 151.1 | - | Kim et al. (2016) [28] |
| Amundsen Sea | 31 December 2013–10 January 2014 | 89.3–991.1 | 69.9–360.5 | 25.4–199.3 | 671.5 ± 311.8 | 3.7 ± 1.6 | Kim et al. (2018) [29] |
| Ross Sea | 25 November 1989–7 January 1990 | 18–279 | 18–650 | 2–94 | 294.4 ± 228.1 | - | Fabiano et al. (1993) [40] |
| Ross Sea (Terra Nova Bay) | austral summer | 32–444 | 108–632 | 3–64 | 374.3 | 1.6 ± 1.3 | Fabiano et al. (1996) [41] |
| Ross Sea (Terra Nova Bay) | February 2015 (ice-free period) | 142.9 ± 55.9 | 143.6 ± 80.5 | 100.3 ± 59.1 | 386.9 ± 194.2 | 2.3 ± 1.2 | Kim et al. (2021) [32] |
| | April–October 2015 (ice-covered period) | 89.0 ± 23.0 | 7.4 ± 7.8 | 23.7 ± 4.6 | 121.1 ± 24.6 | 0.6 ± 0.1 | |
| Chukchi Shelf | 31 July–24 August 2014 | 50.4–480.8 | 25.3–258.5 | 23.7–330.5 | 307.8 ± 284.0 | 1.9 ± 1.8 | This study |
| Canada Basin | | 35.2–90.1 | 2.4–35.1 | 11.7–65.6 | 98.9 ± 26.6 | 0.6 ± 0.2 | This study |

## 5. Conclusions

This study reported the spatial distributions of macromolecular concentrations, compositions, and energy contents of phytoplankton on the Chukchi Shelf and in the Canada Basin. CHO was the major macromolecular component of phytoplankton in the study area, accounting for 41.5% on the Chukchi Shelf and 58.4% in the Canada Basin. The LIP was moderate in both regions. Interestingly, the PRO composition was significantly different between the two regions, showing a low contribution in the Canada Basin (8.7%) and a relatively high contribution on the Chukchi Shelf (27.7%). Severe nutrient-deficient conditions for phytoplankton growth appear to be a major reason for the low PRT composition of phytoplankton in the Canada Basin. In terms of FM concentration and calorie content, a large quantity of high-calorie content food is available in the productive Chukchi Shelf compared to the Canada Basin.

Under the ongoing changes in Arctic environments, the concentration or composition of macromolecules of phytoplankton would be expected to change significantly. Since the different concentrations and compositions determine phytoplankton energy content and consequently regulate the growth and/or nutritional structure of upper trophic levels in the Arctic ecosystem, it is needed to monitor the variability of macromolecular concentration and compositions of phytoplankton. In particular, the macromolecular measurements of phytoplankton with different ocean conditions should be required to better understand how phytoplankton's energy content and its transfer to higher trophic levels are different in the regions of the Arctic Ocean. Recently, using satellite data, Roy et al. [62] tried to estimate the concentrations of CHO, PRT, and LIP and energy values of phytoplankton in the world's oceans. In situ measurement data in various oceans would be useful for improving the global scale estimation based on satellite-derived data.

**Author Contributions:** Conceptualization, K.C., E.Y., J.J. and S.L.; methodology, K.C., M.Y. and S.L.; validation, K.C., M.Y. and S.L.; formal analysis, K.C., S.P., J.K. (Jaejoong Kang), N.J., J.K. (Jaehong Kim), J.K. (Jaesoon Kim) and S.L.; investigation, K.C., E.Y. and J.J.; writing—original draft preparation, K.C.; writing—review and editing, M.Y. and S.L.; visualization, S.P., J.K. (Jaejoong Kang), N.J., J.K. (Jaehong Kim) and J.K. (Jaesoon Kim); supervision, S.L. All authors have read and agreed to the published version of the manuscript.

**Funding:** This research was a part of the project titled "Korea-Arctic Ocean Warming and Responses of Ecosystem (K-AWARE, KOPRI, 1525011760)", funded by the Ministry of Oceans and Fisheries, Korea.

**Institutional Review Board Statement:** Not applicable.

**Informed Consent Statement:** Not applicable.

**Data Availability Statement:** Not applicable.

**Acknowledgments:** We thank the captain and crew of the *ARAON* for their outstanding assistance during the cruise.

**Conflicts of Interest:** The authors declare no conflict of interest.

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
