# Peer review of "Spatial Patterns of Macromolecular Composition of Phytoplankton in the Arctic Ocean"

_water, doi:10.3390/w13182495_

Round 1

Reviewer 1 Report

The study reported the spatial distributions of macromolecular concentrations, compositions, and energy contents of phytoplankton in the Chukchi Shelf and the Canada Basin. The results highlight that the biochemical compositions of phytoplankton are considerably different in the regions of the Arctic Ocean. It is analyzed and predicted that the concentration or composition of macromolecules of phytoplankton would be expected to change significantly under the ongoing changes in Arctic environments. Overall, the results are interesting and the manuscript is relatively well-organized.

Minor comments:

In my opinion, unnecessary abbreviation would decrease the readability of the article. Why the words “lipids” and “proteins” were abbreviated as LIP and PRT?  Are they necessary abbreviation?

Line 64: Please check the grammar of the sentence. The subject was missed.

Line 91-92: “red and blue colors indicate…”?, The colors are not clear for me, in my version ,the Canada Basin are in purple.

Line 95-96: I cannot understand “ different depths (100, 30, and 1% depths of surface PAR).”, what are the specific depths?

Figure 5: some numbers are not clear enough in the figure.

Line 290: Please check the grammar of the sentence. The subject was missed. In my opinion, [27] is not a suitable subject. Similar errors also exist in other lines.

Line 387: “needs” should be “need”.

Line 436: Please check the grammar of the sentence. The subject was missed. In my opinion, [61] is not a suitable subject.  

Author Response

The study reported the spatial distributions of macromolecular concentrations, compositions, and energy contents of phytoplankton in the Chukchi Shelf and the Canada Basin. The results highlight that the biochemical compositions of phytoplankton are considerably different in the regions of the Arctic Ocean. It is analyzed and predicted that the concentration or composition of macromolecules of phytoplankton would be expected to change significantly under the ongoing changes in Arctic environments. Overall, the results are interesting and the manuscript is relatively well-organized.

Minor comments:

In my opinion, unnecessary abbreviation would decrease the readability of the article. Why the words “lipids” and “proteins” were abbreviated as LIP and PRT?  Are they necessary abbreviation?

→ Thank you for your comment. The words “lipids”, “proteins”, and “carbohydrates” are frequently used in this manuscript. Also, the abbreviation was used in previous publications. If it isn’t big deal, we want to keep abbreviations.

Line 64: Please check the grammar of the sentence. The subject was missed.

→ We changed this (in line 63, page 2). Based on the reviewers suggested, we added the author’s name before the number to make it easier to read.

Line 91-92: “red and blue colors indicate…”?, The colors are not clear for me, in my version ,the Canada Basin are in purple.

→ We checked the peer review version again but the Canada Basin was a blue color.

Line 95-96: I cannot understand “different depths (100, 30, and 1% depths of surface PAR).”, what are the specific depths?

→ During water sampling, three photic depths (100%, 30%, and 1% penetration of surface irradiance, PAR) were determined from the underwater PAR sensor (LI-COR underwater optical sensor) lowered with CTD/rosette sampler. Water samples were collected from all of the corresponding light depths and the surface water collected was used for 100% light treatment. To better understand, we added a detailed description in the revised manuscript (in lines 94-97, page 3).

Figure 5: some numbers are not clear enough in the figure.

→ We revised figure 5 to identify clearly.

Line 290: Please check the grammar of the sentence. The subject was missed. In my opinion, [27] is not a suitable subject. Similar errors also exist in other lines.

→ We added the author’s name before the number to make it easier to read (in line 294, page 11). Similar errors in other lines also were changed.

Line 387: “needs” should be “need”.

→ We changed “needs” to “need” (in line 393, page 13).

Line 436: Please check the grammar of the sentence. The subject was missed. In my opinion, [61] is not a suitable subject.

→ We added the author’s name before the number to make it easier to read (in line 442, page 14).

Reviewer 2 Report

1. The article provides an interesting analysis of the spatial patterns of changes in the concentrations of macromolecules of phytoplankton composition in the Arctic Ocean.
However, from the title of the article it follows that the composition of phytoplankton will also be analyzed. But there is no clarification of which composition. I expected this to be the species composition of phytoplankton. But this is not in the article. So the authors can concretize the title of the article, for example - "Spatial patterns of changes in the concentration of macromolecules in the macromolecular composition of phytoplankton in the Arctic Ocean".
2. The article has a conclusion in the "Discussion" section. Maybe it's better to make the conclusion a separate section?
3. In the "References" section, you need to check the design of points 2 and 6

Author Response

  1. The article provides an interesting analysis of the spatial patterns of changes in the concentrations of macromolecules of phytoplankton composition in the Arctic Ocean.
    However, from the title of the article it follows that the composition of phytoplankton will also be analyzed. But there is no clarification of which composition. I expected this to be the species composition of phytoplankton. But this is not in the article. So the authors can concretize the title of the article, for example - "Spatial patterns of changes in the concentration of macromolecules in the macromolecular composition of phytoplankton in the Arctic Ocean".

→ Based on the reviewer suggested, we changed the title into “Spatial patterns of macromolecular composition of phytoplankton in the Arctic Ocean”

  1. The article has a conclusion in the "Discussion" section. Maybe it's better to make the conclusion a separate section?

→ Based on the reviewer suggested, we divided the “Discussion” section into “Discussion” and “Conclusion” sections (in lines 421-445, page 14).

  1. In the "References" section, you need to check the design of points 2 and 6

→ We changed #2 and #6 of Reference (in lines 463 and 470, page 15).

Reviewer 3 Report

In this manuscript, Choe et al. investigated the spatial variance in macromolecule concentration in phytoplankton in the Arctic Ocean. The authors found that macromolecular concentrations varied based on location (shelf vs basin) and with depth. Carbohydrates had the highest concentrations in phytoplankton. This is an interesting study that is well written. I provide some minor comments below that could help improve the manuscript.

General comment

This might be specific to journal format, but I think that just including the citation number (i.e. [13] on line 44) within the sentence makes it difficult to read the sentences. If possible, I would recommend adding the author name before the number to make it easier to read and not have to go to the references to check where the information is referring.

I.e. Kahru et al. [13] and Ardyna et al. [14] for lines 44 and 45.

Line 51: should it be matter instead of matters

Line 272: LIP component accounted for 18.5-62.5% of what?

Line 286: major controlling of what?

Author Response

In this manuscript, Choe et al. investigated the spatial variance in macromolecule concentration in phytoplankton in the Arctic Ocean. The authors found that macromolecular concentrations varied based on location (shelf vs basin) and with depth. Carbohydrates had the highest concentrations in phytoplankton. This is an interesting study that is well written. I provide some minor comments below that could help improve the manuscript.

General comment

This might be specific to journal format, but I think that just including the citation number (i.e. [13] on line 44) within the sentence makes it difficult to read the sentences. If possible, I would recommend adding the author name before the number to make it easier to read and not have to go to the references to check where the information is referring.

I.e. Kahru et al. [13] and Ardyna et al. [14] for lines 44 and 45.

→ Based on the reviewer suggested, we added the author name before the number to make it easier to read (in lines 43,44, page 1). Similar errors in other lines also were changed.

Line 51: should it be matter instead of matters

→ We changed “matters” to “matter” (in line 50, page 2).

Line 272: LIP component accounted for 18.5-62.5% of what?

→ LIP component accounted for 18.5%-62.6% of the total macromolecular composition. We changed this sentence (in line 275, page 10).

Line 286: major controlling of what?

→ This means a major controlling factor affecting the spatial variation of macromolecular compositions. We changed this sentence (in line 290, page 11).

Reviewer 4 Report

General overview:

This paper is focused on some key problems, related to the development of geochemical assessment, distribution, and dynamics of comparison of carbohydrates (CHO), proteins (PRT), lipids (LIP), food material (FM) concentrations, and calorie content of phytoplankton at different regions of the Polar Oceans. Given were range or mean values. The aim of the presented research was to assess the possibilities of spatial distributions of macromolecules.

Specific comments:

Please try to respond to the following issues:

1. Lipid-rich and protein-poor carbon allocation patterns of phytoplankton are important. Please add more citations to the Introduction section.

2. Data choice can be verified by using the instrument of seston lipids in relation to microplankton species composition. Thanks to its use the significance level of particular characteristics can be evaluated, and it may be verified if all selected variables are essential in the next process. Photosynthetic quotients, new production and net community production in the open ocean is currently a top priority due to more intense and more frequent changes in photosynthetic carbon allocation in algal assemblages. How to apply your result in the practice?

3. This study evaluates methods used for photosynthesis quotients, new production and net community production in the open ocean is currently a top priority due to more intense and more frequent changes in photosynthetic carbon allocation in algal assemblages. The aim is to identify an optimal combination of shape factors to measure. The biochemical composition of phytoplankton to estimate long-term temperature change. What is the application of your study for the high incorporation of carbon into proteins by the phytoplankton?

4. The scale of scrutiny of the measurement determines whether or not a temperature changes over time. Combining multiple nutrient stresses and bicarbonate addition to promote lipid accumulation would be helpful. Do you agree with that opinion?

5. Effects of nutrient and light limitation on the biochemical composition of phytoplankton is observed. However, further research is required to determine the performance of the developed method for the seasonality study of seston biochemical composition and the role of carbohydrate accumulation in the growth of planktonic material.

6. This article presents a novel method for the classification of quality, or nutritional value, of phytoplankton. Organic matters produced by phytoplankton have unique values which differ from each other by characteristics such as the synthesis of major macromolecules. The method introduced in this study proposes to use trophic levels. The orientations of edge gradients will be used to analyse the biochemical compositions of phytoplankton under future environmental changes. Data choice can be verified by using the instrument of seston lipids in relation to microplankton species composition. Focusing on the analysis of biological indicators such as a high lipid composition of particulate organic matter and energy content for phytoplankton and physiological conditions is very useful. What do you suggest to improve your results?

7. You have presented the results, both in terms of segmentation and classification, considering a database of nutrients, and implementation of the energy content of Arctic phytoplankton. Which Table or Figure help to understand the transfer of energy that you have pointed in the Introduction section?

Constructive feedback:

Sea ice change, the variability of macromolecular concentration, composition, and calorie content was developed. In this paper, you have presented a system for the Arctic Ocean under ongoing environmental changes. It is based on the composition that predominates as sea ice decreases. Avoiding some classic obstacles of calorie contents of phytoplankton between the two different periods in the Arctic Ocean this approach gives promising results, even on the complex natural phenomenon of an ecological risk index for quantitative concentrations and relative ratios of CHO, PRT, and LIP. The proposed system provides an interactive interface allowing Reader to draw an appropriate conclusion that contains the interesting measurement. Experiments conducted on two distinct databases consisting of seawater samples for nutrient measurements were obtained from different depths (100, 30, and 1% depths of surface PAR) have shown. The appearance of vertical stability and the annual dynamics of nutrients and chlorophyll fluorescence is an important element in the parcel-scale indicators of hydrographic conditions and seasons and provide a critically needed perspective on distributions of phytoplankton carbohydrate assessment studies conducted at much coarser scales — there is usually a requirement that it be consistent and distinct from other studies. The latter aspect is considered in this perspective article with an emphasis on total chlorophyll-a concentration. To realize the full impact of measuring Dissolved inorganic nutrients, major challenges must be addressed. Please expound on this issue. Comparative study of nutrients and chlorophyll-a analysis is needed.

The study also was described the uptake rates of dissolved inorganic carbon and nitrogen by under-ice phytoplankton. The environmental management and temperature, salinity, nutrients, and macromolecular concentration along the shelf/basin gradient are currently important issues for macromolecular compositions and they are major controlling. The system in this study should use a spatial regression model for phytoplankton niches and community composition in the coastal. Statistical analysis of results proceeded in two stages for secular trends in Arctic Ocean net primary production must be modified. The first stage should involve a reduction of variables to those discriminating best, whereas spatial analysis can be made in the second stage. However, your results do not provide any information about the operation of the statistical model on the data obtained in successive years of assessing the suitability of continued increases in Arctic Ocean primary production. Are phytoplankton blooms occurring earlier in the Arctic Sea?

Summary of the research paper:

The purpose of the work was to try to find the role of variation in the macromolecular composition of phytoplankton communities, which will allow their discrimination. Comparison of phytoplankton macromolecular compositions was used for identifying and combining measurements from mobile monitoring and a reference site to develop models of insights on past and future sea-ice evolution from combining observations and models. Therefore, the aim of this study was to develop a spatial distribution of phytoplankton productivity. This paper is focused on some key problems, related to the development of spatial distribution of the macromolecular composition.

Author Response

General overview:

This paper is focused on some key problems, related to the development of geochemical assessment, distribution, and dynamics of comparison of carbohydrates (CHO), proteins (PRT), lipids (LIP), food material (FM) concentrations, and calorie content of phytoplankton at different regions of the Polar Oceans. Given were range or mean values. The aim of the presented research was to assess the possibilities of spatial distributions of macromolecules.

Specific comments:

Please try to respond to the following issues:

  1. Lipid-rich and protein-poor carbon allocation patterns of phytoplankton are important. Please add more citations to the Introduction section.

→ We added more citations to the introduction section (in line 65, page 2).

  1. Data choice can be verified by using the instrument of seston lipids in relation to microplankton species composition. Thanks to its use the significance level of particular characteristics can be evaluated, and it may be verified if all selected variables are essential in the next process. Photosynthetic quotients, new production and net community production in the open ocean is currently a top priority due to more intense and more frequent changes in photosynthetic carbon allocation in algal assemblages. How to apply your result in the practice?

→ Thank you for the reviewer’s comment. As the reviewer mentioned, the change in seston lipid is closely related to microplankton species composition. Similarly, Yun et al. (2019) reported that the change of the species composition could lead the different photosynthetic carbon allocation to macromolecular pools of phytoplankton, mainly related to the lipid synthesis. Since this study is focused on understanding spatial patterns of macromolecular composition and energy content in the phytoplankton community, we didn’t deal with the relationships between species composition and macromolecular composition. We think that this part should be considered in the future study.

→ Under ongoing changes in the ocean environments, the change in photosynthetic carbon allocation and macromolecular composition of phytoplankton could eventually lead to a potential change in total primary production and net community production in the ocean. However, this study could be applied to understanding carbon transfer and/or behavior within and among organisms rather than a quantitative change in primary production in the ocean.

  1. This study evaluates methods used for photosynthesis quotients, new production and net community production in the open ocean is currently a top priority due to more intense and more frequent changes in photosynthetic carbon allocation in algal assemblages. The aim is to identify an optimal combination of shape factors to measure. The biochemical composition of phytoplankton to estimate long-term temperature change. What is the application of your study for the high incorporation of carbon into proteins by the phytoplankton?

→ Temperature change could be one of the important factors affecting macromolecular composition. According to Kim et al. (2021), the proteins concentration during the ice-free period in the Ross Sea of the Antarctic Ocean was 20 times increased than that during the ice-covered period, even though carbohydrates or lipids concentrations showed a slight increase (in lines 381-384, page 13). They concluded that the high proteins concentration during the ice-free period could have resulted from the nutrient increase (reference therein). Even though low water temperatures can reduce the metabolic rate and growth rate of phytoplankton, the temperature was not a major factor in controlling macromolecular concentration and compositions of phytoplankton, especially, in case of the Arctic and Antarctic phytoplankton. Previous studies revealed that temperature fluctuations were related to changes in the fatty acid composition of certain zooplankton (Gladyshev et al., 2010; McMeans et al., 2015). Thus, physiological regulation of microplankton in response to changing water temperature needs to be considered in terms of trophic status and interaction, although temperature change might be insignificant in controlling the macromolecular composition of phytoplankton in this study.

  1. The scale of scrutiny of the measurement determines whether or not a temperature changes over time. Combining multiple nutrient stresses and bicarbonate addition to promote lipid accumulation would be helpful. Do you agree with that opinion?

→ Yes, we also agree with the reviewer’s opinion. Even though this study presented the effect of nutrient limitation on lipid accumulation of phytoplankton, the bicarbonate addition also could be important in understanding lipid accumulation of zooplankton as well as phytoplankton. In case of dissolved organic carbon (DOC), it may be assimilated in the microbial loop and trophically upgraded by heterotrophic protists, thus constituting a potential resource for zooplankton to build lipid reserves or fuel day-to-day metabolism in an environment with low primary production and phytoplankton abundance (Grosbois et al. 2017). Thus, the increasing carbon input under ongoing climate warming might contribute to the lipid reserve accumulation of zooplankton. More studies about combining multiple effects should be required to better understand how multiple factors affect the composition of biochemical pools in lower trophic levels.

  1. Effects of nutrient and light limitation on the biochemical composition of phytoplankton is observed. However, further research is required to determine the performance of the developed method for the seasonality study of seston biochemical composition and the role of carbohydrate accumulation in the growth of planktonic material.

→ Yes, it is important to consider the seasonal variation of the biochemical composition of phytoplankton. In fact, Kim et al. (2021) reported the significant difference in the protein concentration in the Ross Sea of the Antarctic Ocean based on the monthly measurement (in lines 381-384, page, 13). However, in case of Polar Oceans, there are logistic difficulties in measuring the seasonal variation of biochemical composition because of year-round sea-ice presence. Recently, Roy et al. (2018) tried to estimate the concentrations of biochemical components and energy values of phytoplankton, using satellite data (in lines 442-445, page 14). Thus, this method using satellite could be an alternative method for the seasonality study of biochemical composition.

→ Generally, some of carbohydrates serve as food sources and storage, while other ones are cell wall constituents (structural polysaccharides). The role of carbohydrate accumulation in the growth of phytoplankton needs to be better studied since a lot of them have still uncertain functions.

  1. This article presents a novel method for the classification of quality, or nutritional value, of phytoplankton. Organic matters produced by phytoplankton have unique values which differ from each other by characteristics such as the synthesis of major macromolecules. The method introduced in this study proposes to use trophic levels. The orientations of edge gradients will be used to analyse the biochemical compositions of phytoplankton under future environmental changes. Data choice can be verified by using the instrument of seston lipids in relation to microplankton species composition. Focusing on the analysis of biological indicators such as a high lipid composition of particulate organic matter and energy content for phytoplankton and physiological conditions is very useful. What do you suggest to improve your results?

→ Lipids are highly energetic molecules as compared to both carbohydrates and proteins. The change in seston lipid is closely related to microplankton species composition. Indeed, the change of the species composition could lead to the different photosynthetic carbon allocation to macromolecular pools of phytoplankton, mainly related to lipid synthesis (Yun et al., 2019). We need to improve the understanding of the relationship between species composition and macromolecular composition, in particular, regarding lipid composition. In addition, it is essential to better understand how phytoplankton will be affected by ongoing and future environmental changes, and how these changes will drive modifications in the biochemical composition of higher trophic levels in the ecosystem.

  1. You have presented the results, both in terms of segmentation and classification, considering a database of nutrients, and implementation of the energy content of Arctic phytoplankton. Which Table or Figure help to understand the transfer of energy that you have pointed in the Introduction section?

→ We presented the calorie content of phytoplankton in the study area through Tables 1-6. In addition, we compared our values with those at different regions of the Polar Oceans. These values indicate the energy content which can be directly linked to predators (such as microzooplankton, protozoan) through the food web. However, the energy transfer efficiency from prey organisms to the predators could be different with regional or strong seasonal nature. In addition, the energy content of zooplankton is determined by complex ecological parameters (e.g., sex, reproduction, growth stage, and diapause) (Yun et al. 2015). Since this study was here focused on the energy content of phytoplankton as a basic food source, we need to study the energy content of zooplankton together to better understand the energy transfer or interaction between prey (phytoplankton) and predator (zooplankton or protozoan) in the future study.

Constructive feedback:

Sea ice change, the variability of macromolecular concentration, composition, and calorie content was developed. In this paper, you have presented a system for the Arctic Ocean under ongoing environmental changes. It is based on the composition that predominates as sea ice decreases. Avoiding some classic obstacles of calorie contents of phytoplankton between the two different periods in the Arctic Ocean this approach gives promising results, even on the complex natural phenomenon of an ecological risk index for quantitative concentrations and relative ratios of CHO, PRT, and LIP. The proposed system provides an interactive interface allowing Reader to draw an appropriate conclusion that contains the interesting measurement. Experiments conducted on two distinct databases consisting of seawater samples for nutrient measurements were obtained from different depths (100, 30, and 1% depths of surface PAR) have shown. The appearance of vertical stability and the annual dynamics of nutrients and chlorophyll fluorescence is an important element in the parcel-scale indicators of hydrographic conditions and seasons and provide a critically needed perspective on distributions of phytoplankton carbohydrate assessment studies conducted at much coarser scales — there is usually a requirement that it be consistent and distinct from other studies. The latter aspect is considered in this perspective article with an emphasis on total chlorophyll-a concentration. To realize the full impact of measuring Dissolved inorganic nutrients, major challenges must be addressed. Please expound on this issue. Comparative study of nutrients and chlorophyll-a analysis is needed.

→ Thank you for the reviewer’s constructive feedback. The physical structure of the water column and the vertical distribution of chlorophyll and nutrients could be important in understanding the ecological/physiological change of phytoplankton in the ocean. In terms of primary productivity, any changes will be highly localized around bathymetric features or sources of freshwater influx that affect the dynamic balance between stratification and mixing that control nutrient delivery to the euphotic zone. In addition, SCM (Subsurface Chlorophyll Maximum) is well-defined in some peak bloom and most late bloom phases in the Arctic Ocean. According to a previous study, these SCMs were seen to clearly translate into sub-surface peaks in primary production (Hodal and Kristiansen, 2008). However, very little information is available on the physical and chemical structure (such as vertical stability and nutrients dynamics) of the water column and expected biochemical composition variability. Therefore, more field studies are needed to interpret the physical and chemical structure impacts on the vertical distribution of biochemical concentration and composition.

The study also was described the uptake rates of dissolved inorganic carbon and nitrogen by under-ice phytoplankton. The environmental management and temperature, salinity, nutrients, and macromolecular concentration along the shelf/basin gradient are currently important issues for macromolecular compositions and they are major controlling. The system in this study should use a spatial regression model for phytoplankton niches and community composition in the coastal. Statistical analysis of results proceeded in two stages for secular trends in Arctic Ocean net primary production must be modified. The first stage should involve a reduction of variables to those discriminating best, whereas spatial analysis can be made in the second stage. However, your results do not provide any information about the operation of the statistical model on the data obtained in successive years of assessing the suitability of continued increases in Arctic Ocean primary production. Are phytoplankton blooms occurring earlier in the Arctic Sea?

→ As we mentioned above, in situ biological and chemical measurements in the Arctic Ocean (especially, in deep-basin) can be mainly obtained during the summer season because of logistic difficulties. Based on the continuous satellite observations, the decline in sea ice cover and thickness has been reported in the Arctic Ocean. As a result, annual net primary production (NPP) has also been largely increased in ice-free Arctic waters (Arrigo and van Dijken, 2011, 2015; Kahru et al., 2016). In addition, the sea ice changes have considerably affected the bloom timing of phytoplankton (Kahru et al., 2011; Ardyna et al., 2017). An earlier phytoplankton bloom is caused by sea ice loss (Kahru et al., 2011) (in lines 43-44, page 1). The longer open water season in the Arctic has increased the incidence of autumn blooms (Ardyna et al., 2017).

Summary of the research paper:

The purpose of the work was to try to find the role of variation in the macromolecular composition of phytoplankton communities, which will allow their discrimination. Comparison of phytoplankton macromolecular compositions was used for identifying and combining measurements from mobile monitoring and a reference site to develop models of insights on past and future sea-ice evolution from combining observations and models. Therefore, the aim of this study was to develop a spatial distribution of phytoplankton productivity. This paper is focused on some key problems, related to the development of spatial distribution of the macromolecular composition.

Round 2

Reviewer 4 Report

Thanks for the comments. The text and minor errors have been corrected.